# A Nonparametric Weighted Cognitive Diagnosis Model and Its Application on Remedial Instruction in a Small-Class Situation

**Cheng-Hsuan Li \***[ID]**, Yi-Jin Ju and Pei-Jyun Hsieh**

Graduate Institute of Educational Information and Measurement, National Taichung University of Education, Taichung City 40306, Taiwan; jul802001@gmail.com (Y.-J.J.); pp110349@gmail.com (P.-J.H.)
\* Correspondence: chenghsuanli@gmail.com; Tel.: +886-42218-3520

**Abstract:** CDMs can provide a discrete classification of mastery skills to diagnose relevant conceptions immediately for Education Sustainable Development. Due to the problem of parametric CDMs with only a few training sample sizes in small classroom teaching situations and the lack of a nonparametric model for classifying error patterns, two nonparametric weighted cognitive diagnosis models, NWSD and NWBD, for classifying mastery skills and knowledge bugs were proposed, respectively. In both, the variances of items with respect to the ideal responses were considered for computing the weighted Hamming distance, and the inverse distances between the observed and ideal responses were used as weights to obtain the probabilities of the mastering attributes of a student. Conversely, NWBD can classify students' "bugs", so teachers can provide suitable examples for precision assistance before teaching non-mastery skills. According to the experimental results on simulated and real datasets, the proposed methods outperform some standard methods in a small-class situation. The results also demonstrate that a remedial course with NWSD and NWBD is better than one with traditional group remedial teaching.

**Keywords:** nonparametric cognitive diagnosis; skill; misconception; error pattern; small class

## 1. Introduction

Analyzing real-time formative assessments from students to give them relevant and personally recommended learning resources for reducing their cognitive loads is a crucial function of Smart Education (SE) when putting Education Sustainable Development (ESD) into practice [1,2]. In classroom instruction, teachers try to assess what students have learned and identify their strengths and weaknesses to provide appropriate personalized help to each student [3–5]. However, diagnosing students' learning skills and administering different individualized instructions is challenging, especially for a complex teaching skill [6,7]. Additionally, students with some misconceptions may have systematic errors that interfere with their learning [8–11]. This misconception is generated from students' prior learning of numbers: they want to use the same concept to solve the new problem [10–14]. If teachers can identify students' misconceptions and design appropriate feedback, students' later learning can be improved significantly [15].

Many research areas have tried to classify students' strengths and weaknesses. One is based on knowledge space, such as knowledge structure-based adaptive testing [5,16] and meta-knowledge dictionary learning [17]. Another is knowledge tracing models such as deep learning knowledge tracing [18]. The next is cognitive diagnosis models (CDMs), developed to classify the presence and absence of skills or error types. Moreover, CDMs can provide teachers with finer-grain information than unidimensional item response theory [9,19–25]. CDMs are good tools for improving the diagnosis of learning outcomes and have been used in varied applications, such as language assessment [26–29], psychology [24,30], and international examinations [31–33].

Most CDMs are parametric, such as the deterministic input noisy "and" gate (DINA) model, a popular and commonly used CDM. The DINA model uses a slipping parameter and

a guessing parameter of an item to simulate the probabilities of a correct answer [20,21,23,34]. The generalized DINA (G-DINA) model considers the interaction between skills; therefore, it contains more parameters that must be estimated [35]. DINA and G-DINA are robust CDM models and can usually be used for diagnosing mastery skills. In 2021, the G-DINA model framework was applied to investigate primary students' strengths and weaknesses in five digital literacy skills [36]. The bug deterministic input noisy "or" gate (Bug-DINO) model was developed to classify misconceptions [9]. In parametric models, the classification performance relies on estimation methods such as Markov Chain Monte Carlo (MCMC) or expectation maximization (EM) algorithms. The classification performance is also influenced by the sample size [19,37].

A nonparametric cognitive diagnosis model (NPCD) based on the nearest neighbor classifier concept was, therefore, proposed to classify skill mastery patterns. The idea is to classify the observed response vector for a student on a test by finding the closest neighbor among the ideal response patterns determined by the candidates of skill mastery patterns and the Q-matrix of the test. Next, the skill mastery pattern with respect to the ideal response with the closest distance is assigned to the student. Hence, NPCD can be applied to a sample size of 1 without parameter estimation [19] and is more suitable for small-class teaching situations. However, NPCD poses a challenge: more than one ideal response may have the "same and shortest (closest) distance" to the observed response. NPCD randomly selects one of the corresponding candidates of mastery skill patterns and assigns it to the student. In addition, the NPCD with the weighted Hamming distance requires the observed responses of students to estimate variances of items. Hence, it cannot be applied to a small class or just one person for personalized learning.

Therefore, this study proposed a nonparametric weighted skill diagnosis (NWSD) model, which integrates cognitive attribute groups to obtain students' proficiency in various skills and solves the problems encountered by applying NPCD. Furthermore, the variances of ideal responses were applied as weights instead of variances estimated by observed responses of students. This study also extended the NWSD model to the nonparametric weighted bug diagnosis (NWBD) model, which can help teachers diagnose students' error types in small classes.

## 2. NPCD Model

CDMs can be used to provide diagnostic conclusions about examinees' mastery skills. Some are according to a given Q-matrix of a test and their responses [31,34,35,38]. For a test with $J$ items and $K$ attributes with respect to these items, the Q-matrix is as follows:

$$ \mathbf{Q} = \left[ q_{jk} \right]_{j=1,2,\ldots,J,\, k=1,2,\ldots,K} \tag{1} $$

which is a $j \times k$ matrix with each row indicating the required attributes of an item, playing a vital role in CDMs. If the $j$th item required a $k$th attribute, then $q_{jk} = 1$; otherwise, $q_{jk} = 0$ [39,40]. In the conjunctive model, such as the DINA model, students should have all the required attributes of an item, and only then can they answer the item correctly [22,23]. However, in the disjunctive model, such as the deterministic input noisy "or" gate (DINO) model, if students have just one of the required attributes, they have the response 1 of the item. If students have none of the required attributes, they have the response 0 [24]. The idea of a conjunctive model is used to classify mastery skills, so the term "skill" is used instead of the attribute for the conjunctive model.

According to the given Q-matrix of a test, one can have $2^K$ mastery candidate patterns:

$$ \bar{\boldsymbol{\alpha}}_\ell = [\bar{\alpha}_{\ell 1}, \bar{\alpha}_{\ell 2}, \ldots, \bar{\alpha}_{\ell K}], \ \ell = 1, 2, \ldots, 2^K \tag{2} $$

For a student, $\bar{\alpha}_{\ell k} = 1$ indicates the student's mastery of the $k$th skill, and $\bar{\alpha}_{\ell k} = 0$ indicates that the student does not have the $k$th skill. The ideal responses

$$\bar{\boldsymbol{\eta}}_\ell = \left[\bar{\eta}_{\ell 1}, \bar{\eta}_{\ell 2}, \ldots, \bar{\eta}_{\ell J}\right], \ \ell = 1, 2, \ldots, 2^k \tag{3}$$

with respect to the mastery candidate pattern $\bar{\alpha}_\ell$ for a conjunctive model can be computed using

$$\bar{\eta}_{\ell k} = \prod_{k=1}^{K} \bar{\alpha}_{\ell k}^{q_{jk}}, \ \ell = 1, 2, \ldots, 2^k, \ j = 1, 2, \ldots, J. \tag{4}$$

In the NPCD model, the distances between an observed response and the ideal responses are determined according to the Q-matrix [19]. If the observed response of the $i$th student is

$$\boldsymbol{x}_i = \left[x_{i1}, x_{i2}, \ldots, x_{ij}\right], \tag{5}$$

the Hamming distance

$$d_H(\boldsymbol{x}_i, \bar{\boldsymbol{\eta}}_\ell) = \sum_{j=1}^{J} \left| x_{ij} - \bar{\eta}_{\ell j} \right| \tag{6}$$

can be used to calculate the distance between the observed and ideal responses, and then the master pattern $\boldsymbol{\alpha}_i$ is determined according to the master pattern candidate with response to the ideal response whose distance is the minimum, that is,

$$\boldsymbol{\alpha}_i = \bar{\boldsymbol{\alpha}}_{\hat{\ell}}, \ \hat{\ell} = \operatorname*{argmin}_{\ell} d_H(\boldsymbol{x}_i, \bar{\boldsymbol{\eta}}_\ell). \tag{7}$$

However, in some situations, more than one distance between the observed response and the ideal responses is the same, and the distance is the minimum distance:

$$\boldsymbol{\alpha}_i \in \left\{ \bar{\boldsymbol{\alpha}}_{\hat{\ell}_1}, \ \bar{\boldsymbol{\alpha}}_{\hat{\ell}_2}, \ \ldots, \bar{\boldsymbol{\alpha}}_{\hat{\ell}_v} \right\} \tag{8}$$

where $\hat{\ell}_1, \hat{\ell}_2, \ldots, \hat{\ell}_v = \operatorname*{argmin}_{\ell} d_H(\boldsymbol{x}_i, \bar{\boldsymbol{\eta}}_\ell)$. In this case, NPCD randomly selects one of the corresponding mastery pattern candidates for the student. This may reduce the classification accuracy.

Chiu and Douglas proposed a weighted Hamming distance

$$d_{WH}(\boldsymbol{x}_i, \bar{\boldsymbol{\eta}}_\ell) = \sum_{j=1}^{J} \frac{1}{p_j(1 - p_j)} \left| x_{ij} - \bar{\eta}_{\ell j} \right|, \tag{9}$$

where $p_j$ is the correct rate of the $j$th item [19]. Therefore, the larger the variance of an item, the more crucial the corresponding component in the weighted Hamming distance because NPCD tries to identify mastery patterns according to the smallest distance.

Because NPCD only uses distance measures to calculate the similarities, NPCD does not need more students' responses to estimate parameters. Especially if NPCD with the Hamming distance is considered, it is suitable for estimating only one student's mastery pattern. However, if using NPCD with the weighted Hamming distance, the parameter $p_j$ should be estimated according to students' observed responses. Therefore, NPCD with the weighted Hamming distance is also unsuitable for very small classes.

## 3. The Proposed Method: Nonparametric Weighted Cognitive Diagnosis

For applying the weighted Hamming distance in a small class, the variances of the ideal responses are used instead of the variances of the students' observed responses. Moreover, the normalized reciprocals of the weighted Hamming distances are considered as weights to combine the mastery/bug patterns to obtain the probabilities of mastering skills or existing bugs. They are the proposed NWSD and NWBD, respectively—both of which are the nonparametric weighted cognitive diagnosis (NWCD) method.

### 3.1. NWSD Model

The NWSD model based on the concept of expected a posteriori (EAP) probabilities was proposed to solve the problem of NPCD, namely, the fact that some mastery patterns are related to an ideal response. The marginal skill probability of the student $i$ for mastery pattern $\ell$ is also calculated as the sum of all a posteriori $P(\overline{\alpha}_\ell | x_i)$, that is,

$$\widetilde{\alpha}_i = \sum_{\ell=1}^{2^K} P(\overline{\alpha}_\ell | x_i)\overline{\alpha}_\ell, \tag{10}$$

where the a posteriori is estimated according to the normalized inverse distance for the observed response to the ideal responses, that is,

$$P(\overline{\alpha}_\ell | x_i) = \frac{d_{WH}(x_i, \overline{\eta}_\ell)^{-1}}{\sum_{u=1}^{2^K} d_{WH}(x_i, \overline{\eta}_u)^{-1}}. \tag{11}$$

Therefore, the $i$th student's mastery probabilities of skills are estimated as

$$\widetilde{\alpha}_i = [\widetilde{\alpha}_{i1}, \widetilde{\alpha}_{i2}, \ldots, \widetilde{\alpha}_{iK}] = \sum_{\ell=1}^{2^K} \frac{d_{WH}(x_i, \overline{\eta}_\ell)^{-1}}{\sum_{u=1}^{2^K} d_{WH}(x_i, \overline{\eta}_u)^{-1}}\overline{\alpha}_\ell. \tag{12}$$

Note that the same notation of the weighted Hamming distance $d_{WH}(x_i, \overline{\eta}_\ell)$ is used in NWSD, but the weights are calculated according to the variances of the ideal responses, that is,

$$d_{WH}(x_i, \overline{\eta}_\ell) = \sum_{j=1}^{J} \frac{1}{\overline{p}_j\left(1 - \overline{p}_j\right)}\left|x_{ij} - \overline{\eta}_{\ell j}\right|, \tag{13}$$

where

$$\overline{p}_j = \frac{\sum_{\ell=1}^{2^K} \overline{\eta}_{\ell j}}{2^k}, \ j = 1, 2, \ldots, J. \tag{14}$$

For deducing the discrete skill class, if the $k$th skill probability ($k$th component of $\alpha_i = [\alpha_{i1}, \alpha_{i2}, \ldots, \alpha_{iK}]$) is greater than or equal to a given threshold $\varepsilon_s$, then NWSD classifies that the examinee has mastered the $k$th skill. Otherwise, if the $k$th skill probability is smaller than $\varepsilon_s$, then NWSD classifies that the examinee has not mastered the $k$th skill. Therefore,

$$\alpha_{ik} = \begin{cases} 1 & if \ \widetilde{\alpha}_{ik} > \varepsilon_s \\ 0 & if \ \widetilde{\alpha}_{ik} \leq \varepsilon_s \end{cases}. \tag{15}$$

In addition, if only the smallest distance from the observed response and ideal responses is considered, the smallest distance is set to 1, and the other distances are set to 0, then the proposed NWSD degenerates to NPCD. A commonly used threshold for CDMs is 0.5 [3,41,42].

### 3.2. NWBD Model

The idea of a disjunctive model is used to classify existing bugs or misconceptions, and hence, the term "bug" is used to indicate a student has error types or misconceptions instead of the attribute for the disjunctive model. Moreover, the term "M-matrix ($M = [m_{jr}]$, an $J \times R$ matrix)" is used instead of "Q-matrix" for ease of understanding. For NWBD, the corresponding bug patterns are

$$\overline{\beta}_t = [\overline{\beta}_{t1}, \overline{\beta}_{t2}, \ldots, \overline{\beta}_{tR}], t = 1, 2, \ldots, 2^R. \tag{16}$$

Moreover, the bug ideal responses are

$$\overline{\gamma}_t = [\overline{\gamma}_{t1}, \overline{\gamma}_{t2}, \ldots, \overline{\gamma}_{tJ}], t = 1, 2, \ldots, 2^R, \tag{17}$$

where

$$\overline{\gamma}_{tj} = \prod_{r=1}^{R} (1 - \overline{\beta}_{tr})^{m_{jr}}. \tag{18}$$

Note that if a student whose bug pattern is $\boldsymbol{\beta}_i \in \{\overline{\boldsymbol{\beta}}_t\}_{t=1,2,\ldots,2^R}$ has at least one misconception of the *j*th item, then $\gamma_{ij} = 0$, where

$$\gamma_{ij} = \prod_{r=1}^{R} (1 - \beta_{ir})^{m_{jr}}. \tag{19}$$

Otherwise, if a student does not have any bugs with the *j*th item, then $\gamma_{ij} = 1$. Similar to NWSD, the weights of the weighted Hamming distance are calculated using the variances of the bug ideal responses:

$$d_{WH}(\boldsymbol{x}_i, \overline{\boldsymbol{\gamma}}_t) = \sum_{j=1}^{J} \frac{1}{\overline{p}_j(1 - \overline{p}_j)} |x_{ij} - \overline{\gamma}_{tj}|, \tag{20}$$

where

$$\overline{p}_j = \frac{\sum_{t=1}^{2^R} \overline{\gamma}_{tj}}{2^k}, \ j = 1, 2, \ldots, J. \tag{21}$$

Finally, the a posteriori of the EAP,

$$P(\overline{\boldsymbol{\beta}}_t | \boldsymbol{x}_i) = \frac{d_{WH}(\boldsymbol{x}_i, \overline{\boldsymbol{\gamma}}_t)^{-1}}{\sum_{v=1}^{2^R} d_{WH}(\boldsymbol{x}_i, \overline{\boldsymbol{\gamma}}_v)^{-1}}, \tag{22}$$

is used to integrate the bug candidate patterns $\overline{\boldsymbol{\beta}}_t, t = 1, 2, \ldots, 2^R$, and the probabilities of bugs of the student, $\widetilde{\beta}_{i1}, \widetilde{\beta}_{i2}, \ldots, \widetilde{\beta}_{iR}$, can be obtained by

$$\widetilde{\boldsymbol{\beta}}_i = \left[\widetilde{\beta}_{i1}, \widetilde{\beta}_{i2}, \ldots, \widetilde{\beta}_{iR}\right] = \sum_{t=1}^{2^R} \frac{d_{WH}(\boldsymbol{x}_i, \overline{\boldsymbol{\gamma}}_t)^{-1}}{\sum_{v=1}^{2^R} d_{WH}(\boldsymbol{x}_i, \overline{\boldsymbol{\gamma}}_v)^{-1}} \overline{\boldsymbol{\beta}}_t. \tag{23}$$

If $\widetilde{\beta}_{ir}$ is greater than a given threshold $\varepsilon_b$, then the estimated bug $\beta_{ir} = 1$ in the bug pattern $\boldsymbol{\beta}_i = [\beta_{i1}, \beta_{i2}, \ldots, \beta_{iR}]$.

### 3.3. A Nonparametric CDM Website

A web graphical user interface (GUI) was developed using R shiny [43] and can be found at https://chenghsuanli.shinyapps.io/NPWCD/ (accessed on 18 March 2022). Figure 1 presents the diagnostic results of examinees by applying NWSD from the website. Teachers can obtain the list of mastery skills through the following four steps by the chosen nonparametric CDMs.

1. Upload the Q-matrix saved in a CSV (Comma-Separated Values) file.
2. Upload students' responses saved in a CSV file.
3. Choose the nonparametric CDMs, NPCD, NWSD, or NWBD from the method list.
4. Press the "Go" button to obtain the mastery profiles of students.

From Figure 1, teachers can find the first six rows of the Q-matrix and responses to help them check the data. Moreover, the final mastery profile will be shown at the bottom of the right panel after they press the "Go" button. If teachers want to save the mastery profile, they can press the "Download" bottom to keep the mastery profile in a CSV file.

**Figure 1.** Diagnostic results of examinees by applying NWSD from the website.

## 4. Simulation Studies on Artificial and Real Datasets

Two cases of simulated datasets were used to verify the proposed methods, NWSD and NWBD, in the small-class situation. Case 1 was generated according to the simultaneously identifying skills and misconceptions (SISM) model [37]. The other (Case 2) was randomly selected observed responses from a real dataset [37,44,45] to form small-class datasets.

Two classification agreement rates, the pattern-wise agreement rate (PAR),

$$\text{PAR} = \frac{1}{I} \sum_{i=1}^{I} \prod_{k=1}^{K} \delta(\alpha_{ik}, \hat{\alpha}_{ik}) \tag{24}$$

for classifying skills, and

$$\text{PAR} = \frac{1}{I} \sum_{i=1}^{I} \prod_{r=1}^{R} \delta(\beta_{ir}, \hat{\beta}_{ir}) \tag{25}$$

for classifying bugs, and the attribute-wise agreement rate (AAR),

$$\text{AAR} = \frac{1}{I \times K} \sum_{i=1}^{I} \sum_{k=1}^{K} \delta(\alpha_{ik}, \hat{\alpha}_{ik}) \tag{26}$$

for classifying skills, and

$$\text{AAR} = \frac{1}{I \times R} \sum_{i=1}^{I} \sum_{r=1}^{R} \delta(\beta_{ir}, \hat{\beta}_{ir}) \tag{27}$$

for classifying bugs, were computed for comparison, where $I$ indicates the number of students;

$$\delta(a, b) = \begin{cases} 1 & if\ a = b \\ 0 & if\ a \neq b \end{cases}; \tag{28}$$

$\alpha_{ik}$ and $\beta_{ir}$ for $k = 1, 2, \ldots, K$, $r = 1, 2, \ldots, R$ indicate the true attributes (skill or bug) of the $i$th student, respectively; $\hat{\alpha}_{ik}$ and $\hat{\beta}_{ir}$ for $k = 1, 2, \ldots, K$, $r = 1, 2, \ldots, R$ indicate the estimated attributes from the CDM model of the $i$th student, respectively. The mean and the standard deviation of 500 replications of AARs and PARs were computed for the comparison [44].

*4.1. Case 1*

In Case 1, the simulated dataset was generated from SISM according to the Q-matrix, the M-matrix presented in Table 1, and the following parameter setting: $h = 0.95$, $\omega = 0.15$, $g = 0.35$, and $\epsilon = 0.05$. These parameters were related to (a) the success probability of students who have mastered all skills and possess no bugs, (b) the success probability of students who have mastered all skills but possess at least one bug, (c) the success probability of students who have not mastered all skills and possess no bugs, and (d) the success probability of students who have not mastered all skills and possess at least one bug, respectively [37,46,47]. These settings were related to the high-quality items because the success probability of students who mastered all skills and possessed no bugs ($h$) was higher, and the remaining parameters ($\omega$, $g$, and $\epsilon$) were quite lower values. The simulated test included 17 items, 4 required skills (S1–S4), and 3 existing bugs (B1–B3). To discuss the small-class situation, three sample sizes ($I$ = 20, 50, and 100) of students were generated and analyzed.

**Table 1.** Q-matrix and M-matrix of Case 1.

| Item | Q-Matrix (w.r.t. Skills) | | | | M-Matrix (w.r.t. Bugs) | | |
|:---:|:---:|:---:|:---:|:---:|:---:|:---:|:---:|
| | S1 | S2 | S3 | S4 | B1 | B2 | B3 |
| 1 | 1 | 0 | 0 | 0 | 0 | 0 | 0 |
| 2 | 0 | 1 | 0 | 0 | 0 | 0 | 0 |
| 3 | 0 | 0 | 1 | 0 | 0 | 0 | 0 |
| 4 | 0 | 0 | 0 | 1 | 0 | 0 | 0 |
| 5 | 1 | 0 | 0 | 0 | 1 | 0 | 0 |
| 6 | 0 | 1 | 0 | 0 | 1 | 0 | 0 |
| 7 | 0 | 0 | 1 | 0 | 0 | 0 | 1 |
| 8 | 0 | 0 | 0 | 1 | 0 | 1 | 0 |
| 9 | 1 | 1 | 0 | 0 | 1 | 0 | 0 |
| 10 | 1 | 0 | 1 | 0 | 0 | 0 | 1 |
| 11 | 1 | 0 | 0 | 1 | 0 | 0 | 1 |
| 12 | 0 | 1 | 1 | 0 | 0 | 0 | 1 |
| 13 | 0 | 1 | 0 | 1 | 0 | 1 | 1 |
| 14 | 0 | 0 | 1 | 1 | 0 | 1 | 1 |
| 15 | 1 | 0 | 1 | 0 | 1 | 1 | 0 |
| 16 | 1 | 1 | 0 | 1 | 1 | 1 | 0 |
| 17 | 0 | 1 | 1 | 1 | 1 | 1 | 0 |

Tables 2 and 3 present the average classification agreements for skills and bugs, respectively. For classifying skills, no matter the model, the classification agreements were consistent among $I$ = 20, 50, and 100. Moreover, AARs and PARs were above 0.8100 and 0.4500, respectively. The highest average AARs and PARs among $I$ = 20, 50, and 100 were obtained from the proposed NWSD. They were >0.8400 and >0.5100, respectively. Similar results are presented in Table 3 for classifying bugs. The average classification agreements of AARs and PARs of NWBD were higher than those of Bug-DINO. In addition, they were >0.7800 and >0.4800, respectively.

**Table 2.** Average AARs and PARs of DINA, G-DINA, NPCD, and NWSD models in Case 1 (corresponding standard deviations are shown in parentheses).

| Model | Classification Agreement Rate | *I* = 20 | *I* = 50 | *I* = 100 |
|---|---|---|---|---|
| DINA | AAR | 0.8137 (0.06) | 0.8194 (0.04) | 0.8281 (0.03) |
| | PAR | 0.4588 (0.13) | 0.4673 (0.09) | 0.4812 (0.07) |
| G-DINA | AAR | 0.8248 (0.05) | 0.8195 (0.03) | 0.8225 (0.03) |
| | PAR | 0.4741 (0.12) | 0.4591 (0.08) | 0.4624 (0.07) |
| NPCD | AAR | 0.8297 (0.04) | 0.8337 (0.03) | 0.8328 (0.02) |
| | PAR | 0.4965 (0.11) | 0.5023 (0.07) | 0.4981 (0.05) |
| NWSD | AAR | 0.8437 (0.05) | 0.8496 (0.03) | 0.8509 (0.02) |
| | PAR | 0.5130 (0.12) | 0.5243 (0.07) | 0.5253 (0.05) |

**Table 3.** Average AARs and PARs of Bug-DINO and NWBD models in Case 1 (corresponding standard deviations are shown in parentheses).

| Model | Classification Agreement Rate | *I* = 20 | *I* = 50 | *I* = 100 |
|---|---|---|---|---|
| Bug-DINO | AAR | 0.7158 (0.06) | 0.7233 (0.05) | 0.7215 (0.03) |
| | PAR | 0.3683 (0.10) | 0.3825 (0.07) | 0.3799 (0.05) |
| NWBD | AAR | 0.7818 (0.06) | 0.7966 (0.05) | 0.8010 (0.03) |
| | PAR | 0.4850 (0.12) | 0.5121 (0.08) | 0.5201 (0.05) |

*4.2. Case 2*

Real-world data, the fraction multiplication data, were applied to verify the proposed NWCD model method to demonstrate the real-world application of our CDM [37,44,45]. There were 286 grade 3 students from elementary schools in Taiwan participating in the test. This test has seven open-ended fraction multiplication items. Four skills for diagnosis were considered in the test: S1 (the ability to multiply a whole number by a fraction), S2 (the ability to multiply a fraction by a fraction), S3 (the ability to reduce the answer to its lowest terms), and S4 (the ability to solve a two-step problem). Moreover, the experts indicated three bugs: B1 (turning the second fraction upside down when multiplying a fraction by a fraction), B2 (solving only the first step of a two-step problem), and B3 (performing incorrect arithmetic operations when confused about the relational terms). The Q-matrix with respect to these misconceptions is presented in Table 4.

**Table 4.** Q-matrix and M-matrix of Case 1.

| Item | Q-Matrix (w.r.t. Skills) | | | | M-Matrix (w.r.t. Bugs) | | |
|---|---|---|---|---|---|---|---|
| | **S1** | **S2** | **S3** | **S4** | **B1** | **B2** | **B3** |
| 1 | 1 | 1 | 0 | 1 | 1 | 1 | 0 |
| 2 | 0 | 1 | 1 | 0 | 1 | 0 | 0 |
| 3 | 0 | 1 | 0 | 1 | 0 | 1 | 1 |
| 4 | 1 | 1 | 0 | 1 | 1 | 1 | 0 |
| 5 | 1 | 1 | 1 | 1 | 1 | 1 | 0 |
| 6 | 1 | 1 | 0 | 1 | 1 | 1 | 0 |
| 7 | 0 | 1 | 1 | 1 | 0 | 1 | 0 |

Students who participated in the test were required to write down their problem-solving processes of items in detail while selecting a choice of answers. Moreover, a group of experts identified students' latent skills and existing bugs according to their recorded problem-solving processes. These were the benchmarks of students for comparison [37,44,45]. For simulating small-class teaching, similar to Case 1, three sample sizes, $I = 20$, 50, and 100, were randomly selected from 286 students in the whole dataset.

Tables 5 and 6 present the average classification agreements for skills and bugs, respectively. For classifying skills, no matter the model, the classification agreements were consistent among $I = 20$, 50, and 100. Moreover, AARs and PARs of DINA, NPCD, and NWSD were >0.7700 and >0.4700, respectively. The highest average AARs and PARs among $I = 20$, 50, and 100 were obtained from the proposed NWSD. They were >0.8300 and >0.6400, respectively. The average classification agreements were a little poor and may be influenced by the test design (i.e., the type of Q-matrix). Similar results are presented in Table 6 for classifying bugs. The average classification agreements of AARs and PARs of NWBD are >0.7680 and >0.4200, respectively, which are higher than those of Bug-DINO.

**Table 5.** Average AARs and PARs of DINA, G-DINA, NPCD, and NWSD models in Case 2 (corresponding standard deviations are shown in parentheses).

| Model | Classification Agreement Rate | $I = 20$ | $I = 50$ | $I = 100$ |
|---|---|---|---|---|
| DINA | AAR | 0.7716 (0.06) | 0.7797 (0.04) | 0.7774 (0.03) |
| | PAR | 0.4776 (0.13) | 0.4849 (0.10) | 0.4783 (0.07) |
| G-DINA | AAR | 0.5438 (0.07) | 0.5498 (0.07) | 0.5505 (0.07) |
| | PAR | 0.0814 (0.08) | 0.1026 (0.08) | 0.1138 (0.09) |
| NPCD | AAR | 0.7894 (0.06) | 0.7893 (0.03) | 0.7875 (0.02) |
| | PAR | 0.5370 (0.12) | 0.5368 (0.06) | 0.5366 (0.04) |
| NWSD | AAR | 0.8387 (0.07) | 0.8408 (0.03) | 0.8392 (0.02) |
| | PAR | 0.6495 (0.14) | 0.6532 (0.07) | 0.6483 (0.04) |

**Table 6.** Average AARs and PARs of Bug-DINO and NWBD models in Case 2 (corresponding standard deviations are shown in parentheses).

| Model | Classification Agreement Rate | *I* = 20 | *I* = 50 | *I* = 100 |
|---|---|---|---|---|
| Bug-DINO | AAR | 0.7161 (0.06) | 0.7162 (0.03) | 0.7161 (0.02) |
| | PAR | 0.3972 (0.10) | 0.3923 (0.06) | 0.3932 (0.04) |
| NWBD | AAR | 0.7695 (0.06) | 0.7684 (0.03) | 0.7680 (0.02) |
| | PAR | 0.4422 (0.12) | 0.4311 (0.07) | 0.4292 (0.05) |

## 5. Experiments on Remedial Instruction

Two experiments were conducted on remedial instruction. One considered only the required skills for a test, and the other considered both required skills and existing bugs, which requires teachers' experiences in teaching based on which they can indicate existing bugs for a given test. Two remedial instruction groups, the personalized instruction group according to the report by the NWCD model (experimental group) and traditional group remedial teaching (control group), were considered.

### 5.1. Personalized Instruction Based on NWSD

Ninety-four eighth-grade students from a Taiwanese junior high school participated in this study. The students were from 2 classes, and each had 47 students. One class was randomly selected as the experimental group and the other as the control group. All participants attended the high school mathematics class, "Series and Arithmetic Series". The same curriculum was used for both groups.

The pretest and posttest (20 items each) were designed based on five skills (Table 7). Thus, both have the same Q-matrix (Table 8). In the experimental group, the personalized mastery patterns of individuals were provided according to NWSD with the dichotomous responses and Q-matrix. Moreover, individuals learned through videos from the "Adaptive Learning Platform" [48] according to their lack of skills. The period was 40 min. Note that individuals' learning time may be different because the lack of skills may be different. In the control group, the traditional group remedial instruction based on the result of the pretest was performed by the teacher in 40 min (Figure 2). The control factors were eighth-grade students, instruction time, pretest time, and posttest time.

**Table 7.** Five skills from the mathematical topic, "Series and Arithmetic Series".

| Skill | Description |
|---|---|
| S1 | Understanding the meaning of a series |
| S2 | Understanding the meaning of an arithmetic series |
| S3 | Calculate the sum of a finite arithmetic series |
| S4 | Understanding and applying the formulation of the sum of a finite arithmetic series |
| S5 | Applying the formulation of the sum of a finite arithmetic series to a real-world problem |

**Table 8.** Q-matrix for the 20-item test assessing five skills.

| Item | S1 | S2 | S3 | S4 | S5 | Item | S1 | S2 | S3 | S4 | S5 |
|---|---|---|---|---|---|---|---|---|---|---|---|
| 1 | 1 | 0 | 0 | 0 | 0 | 11 | 0 | 1 | 1 | 0 | 0 |
| 2 | 0 | 1 | 0 | 0 | 0 | 12 | 0 | 1 | 0 | 1 | 0 |
| 3 | 0 | 1 | 0 | 0 | 0 | 13 | 0 | 1 | 0 | 0 | 1 |
| 4 | 0 | 0 | 1 | 0 | 0 | 14 | 0 | 0 | 1 | 1 | 0 |
| 5 | 0 | 0 | 0 | 1 | 0 | 15 | 0 | 0 | 1 | 0 | 1 |

**Table 8.** *Cont.*

| Item | S1 | S2 | S3 | S4 | S5 | Item | S1 | S2 | S3 | S4 | S5 |
|------|----|----|----|----|----|------|----|----|----|----|----|
| 6 | 0 | 0 | 0 | 0 | 1 | 16 | 1 | 1 | 1 | 0 | 0 |
| 7 | 1 | 1 | 0 | 0 | 0 | 17 | 1 | 1 | 0 | 1 | 0 |
| 8 | 1 | 0 | 1 | 0 | 0 | 18 | 1 | 1 | 0 | 0 | 1 |
| 9 | 1 | 0 | 0 | 1 | 0 | 19 | 1 | 0 | 1 | 0 | 1 |
| 10 | 1 | 0 | 0 | 0 | 1 | 20 | 1 | 1 | 1 | 1 | 1 |

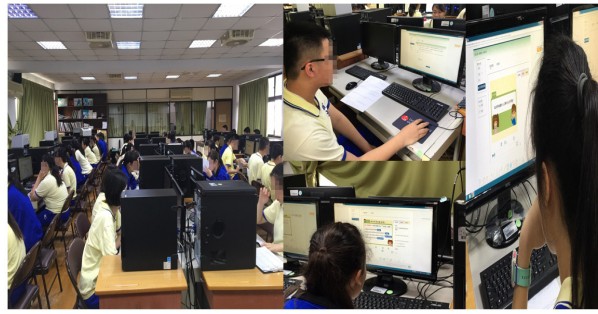 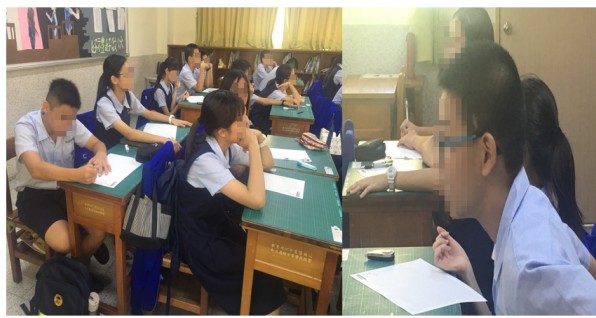

(**a**) Experimental Group      (**b**) Control Group

**Figure 2.** The remedial instructions for (**a**) Experimental Group with individual video instruction and (**b**) Control Group with traditional group instruction.

To investigate whether students improved after participating in the remedial instructions, a paired sample *t*-test was applied, and the results are shown in Table 9. The differences in mean scores between pretest and posttest were 27.021 and 3.255 for the individual remedial instruction by teaching videos from the "Adaptive Learning Platform" based on the reports from NWSD (the experimental group) and for the traditional group teaching (control group), respectively. Table 9 also shows that the average score of the posttest was statistically significantly higher than the average pretest score for both groups. Hence, the students who participated in both remedial instructions showed an improvement in their learning performance.

**Table 9.** Results of difference in scores between pretest and posttest for three groups.

| Group | Mean Pretest Score | Mean Post-Test Score | *t*-Value |
|-------|-------------------|----------------------|-----------|
| Experimental Group | 46.596 | 73.617 | 14.598 *** |
| Control Group | 66.851 | 70.105 | 2.295 * |

Note. * $p < 0.05$, *** $p < 0.001$.

The differences in learning effectiveness of the different remedial instructions were compared using an analysis of covariance. The homogeneity of variance assumes that both groups had acceptable and equal error variances ($F = 1.475$; $p = 0.228$), as determined using Levene's test. Moreover, the homogeneity for regression coefficients within both groups was confirmed because the assumption for homogeneity of regression coefficients was also conducted ($F = 2.296$; $p = 0.133$).

Table 10 presents the results of ANCOVA, demonstrating the effect of two remedial instructions on posttest scores after adjusting for the effect of the pretest scores. A significant difference is noted in posttest scores between the two groups ($F = 54.960$ ***). The results of Fisher's least significant difference (LSD) reveal that the experimental group significantly outperformed the control group (13.658, $p < 0.001$) because individuals received personalized learning videos according to their lack of skills determined by the proposed NWSD.

**Table 10.** Results of ANCOVA on the learning effectiveness of the two remedial instructions.

| Variable | Level | Mean [a] (SE) | *F* Values | Post Hoc [b] |
|---|---|---|---|---|
| Pretest | | | 112.031 *** | |
| DTRIP | Experimental Group | 78.691 | 54.960 *** | Experimental Group > Control Group *** |
| | Control Group | 65.032 | | |

Note. *** $p < 0.001$. DTRIP: Different types of classes. [a] = Covariates appearing in the model are evaluated at the following value: 56.72. [b] = Adjustment for multiple comparisons: least significant difference (equivalent to no adjustments).

*5.2. Personalized Instruction Based on NWSD and NWBD*

Sixty-seven students from the ninth grade of a Taiwanese junior high school participated in this study. The experimental and control groups had 32 and 35 students, respectively. All participants attended the high school science class, "Rectilinear Motion". The curriculum was the same for both groups. The course was divided into five subunits. In each unit, students in both groups attended a pretest and a posttest, which were designed according to the same Q-matrix and M-matrix. For example, the fourth unit, "Uniform Accelerated Motion", had seven skills and five bugs (Table 11), and the corresponding Q-matrix and M-matrix are presented in Table 12.

**Table 11.** Seven skills and five bugs based on the unit "Uniform Accelerated Motion".

| Type | Name | Description |
|---|---|---|
| Skill | S1 | Understanding the definition of average acceleration |
| | S2 | Understanding the conversion between speed units |
| | S3 | Understanding that the speed of the object will change when the object moves with acceleration |
| | S4 | Understanding the change of speed when the directions of speed and acceleration change |
| | S5 | Understanding the V-t diagram of constant acceleration motion is an oblique straight line |
| | S6 | Judging the direction of acceleration by the V-t diagram |
| | S7 | Understanding the area enclosed by the V-t diagram and the time axis represents "displacement" |
| Bugs | B1 | Calculating speed change by using large speed and small speed |
| | B2 | Calculating the average acceleration by using the speed on the V-t diagram divided by the time |
| | B3 | If the acceleration is a positive (negative) value, then the object will increase (decrease) speed |
| | B4 | The acceleration is a positive value when the V-t diagram appears in the first quadrant.The acceleration is a negative value when the V-t diagram appears in the fourth quadrant. |
| | B5 | If the figure is drawn up (down), then the displacement direction is the positive (negative) direction. |

In the experimental group, the personalized mastery patterns, including individual skills and bugs, were provided according to NWSD and NWBD with the dichotomous responses, Q-matrix, and M-matrix. The teacher used appropriate cognitive conflict strategies to clarify students' misconceptions, according to the bug reports of NWBD. Moreover, students were taught the skills they lacked, according to the report of NWSD. In the control group, the traditional group remedial instruction based on the results of the pretest was performed by the teacher.

**Table 12.** Q-matrix and M-matrix of the unit "Uniform Accelerated Motion".

| Item | Q-Matrix (w.r.t. Skills) | | | | | | | M-Matrix (w.r.t. Bugs) | | | | |
|---|---|---|---|---|---|---|---|---|---|---|---|---|
| | S1 | S2 | S3 | S4 | S5 | S6 | S7 | B1 | B2 | B3 | B4 | B5 |
| 1 | 1 | 0 | 0 | 0 | 0 | 0 | 0 | 1 | 0 | 0 | 0 | 0 |
| 2 | 1 | 1 | 0 | 0 | 0 | 0 | 0 | 1 | 0 | 0 | 0 | 0 |
| 3 | 1 | 0 | 0 | 0 | 0 | 0 | 0 | 0 | 1 | 0 | 0 | 0 |
| 4 | 1 | 0 | 0 | 0 | 0 | 0 | 0 | 1 | 1 | 0 | 0 | 0 |
| 5 | 1 | 0 | 0 | 0 | 0 | 0 | 0 | 0 | 0 | 0 | 0 | 0 |
| 6 | 0 | 0 | 1 | 0 | 0 | 0 | 0 | 0 | 0 | 0 | 0 | 0 |
| 7 | 0 | 0 | 1 | 0 | 0 | 0 | 0 | 0 | 0 | 0 | 0 | 0 |
| 8 | 0 | 0 | 0 | 1 | 0 | 0 | 0 | 0 | 0 | 1 | 0 | 0 |
| 9 | 0 | 0 | 0 | 0 | 1 | 0 | 0 | 0 | 0 | 0 | 0 | 0 |
| 10 | 0 | 0 | 0 | 0 | 0 | 1 | 0 | 0 | 0 | 0 | 1 | 0 |
| 11 | 0 | 0 | 0 | 1 | 0 | 1 | 0 | 0 | 0 | 0 | 0 | 0 |
| 12 | 0 | 0 | 0 | 0 | 0 | 0 | 1 | 0 | 0 | 0 | 0 | 0 |
| 13 | 0 | 0 | 0 | 0 | 0 | 0 | 1 | 0 | 0 | 0 | 0 | 0 |
| 14 | 0 | 0 | 0 | 1 | 0 | 1 | 1 | 0 | 0 | 0 | 0 | 0 |
| 15 | 0 | 0 | 0 | 1 | 1 | 1 | 0 | 0 | 0 | 0 | 1 | 0 |
| 16 | 0 | 0 | 0 | 0 | 1 | 0 | 1 | 0 | 0 | 0 | 0 | 0 |
| 17 | 0 | 0 | 0 | 1 | 0 | 1 | 1 | 0 | 0 | 0 | 0 | 1 |
| 18 | 0 | 0 | 0 | 1 | 1 | 1 | 1 | 0 | 0 | 0 | 0 | 1 |
| 19 | 0 | 0 | 0 | 0 | 0 | 1 | 1 | 0 | 0 | 0 | 1 | 0 |
| 20 | 0 | 0 | 0 | 0 | 1 | 0 | 1 | 0 | 0 | 0 | 0 | 0 |

The differences in learning effectiveness of the different remedial instructions were compared using ANCOVA. The scores from a test before they participated in the course were considered the covariates, and the scores of the final posttest of the fifth unit were regarded as the dependent variables. The homogeneity for regression coefficients within both groups was confirmed because the assumption for homogeneity of regression coefficients was also conducted ($F = 0.608$; $p = 0.438$). Table 13 presents the ANCOVA results, demonstrating the effect of two remedial instructions on the posttest scores according to adjusting for the effect of the covariate. The difference in posttest scores between the groups was significant ($F = 11.965$ **). Fisher's LSD test indicated that the experimental group significantly outperformed the control group (12.309, $p < 0.01$).

**Table 13.** Results of ANCOVA on the learning effectiveness of the two remedial instructions.

| Variable | Level | Mean (SE) | F Values | Post Hoc [b] |
|---|---|---|---|---|
| Covariate | | | 8.429 ** | |
| DTRIP | Experimental Group | 87.833 | 11.965 ** | Experimental Group > Control Group ** |
| | Control Group | 75.524 | | |

Note. ** $p < 0.01$. DTRIP: Different types of classes. [b] = Adjustment for multiple comparisons: the least significant difference (equivalent to no adjustments).

## 6. Discussion

Some pre-service or in-service teachers seem insufficient to diagnose students' conceptions, an essential and challenging task for ESD implementation [49]. This study extended NPCD as NWCDs for obtaining more accurate individual profiles of mastery skills and error types of students to achieve the diagnostic phase of precision education [50,51]. The proposed methods attempted to integrate the ideal responses by weights determined according to the distances between a student's observed response and ideal responses. Thus, NWCDs do not need observed responses to find suitable parameters, which is the problem in applying parametric CDM models, such as DINA and G-DINA, to small-class teaching. Therefore, both NWSD and NWBD can be applied to small-class teaching (approximately 30 students in a class in Taiwan) or only one student. NWCDs with the "Taiwan Adaptive

Learning Platform" [48] can provide an intelligent and personalized adaptive learning environment to commit to ESD [1,2].

The results of the simulation studies of datasets with <100 examinees generated from both SISM (an artificial dataset) and a real dataset indicate that both NWSD and NWBD have the best classification agreement compared with that of traditional nonparametric CDM, NPCD, and the two parametric CDMs (DINA and G-DINA). Therefore, NWCD works well in small-class teaching. Moreover, the report from NWCD can show both mastery skills and error types. Teachers can provide individual feedback and teaching materials, such as individual teaching videos and worksheets, instead of traditional group remedial teaching (non-personalized instruction). It can increase not only learning performance but also the efficiency of classroom instructional time [52].

For class teaching, NWSD has been applied to provide appropriate learning videos according to the students' lack of skills. The results based on the pretest and posttest of students who attended the assessment of the mathematics unit, "Series and Arithmetic Series", in a junior high school in Taiwan show that the corresponding remedial class with personalized instruction is significantly better than one with traditional group remedial instruction by the teacher. If teachers have more teaching experience, they can also consider students' bugs or misconceptions. This can provide extra information to test and form the M-matrix. Based on both the M-matrix and Q-matrix, NWBD and NWSD can be applied to classify existing bugs and the lack of skills, respectively. The experimental results for the science topic, "Rectilinear Motion" for junior high school students in Taiwan show that if the students have been taught by cognitive conflict methods based on their existing bugs/misconceptions first and then taught their lack of skills, the corresponding average learning performance was higher than that of students who were taught using traditional group remedial instruction.

## 7. Conclusions

For Sustainable Development Goals 4, helping teachers understand students' mastery skills and error types is essential in SE for ESD implementation. Chiu and Douglas [19] proposed NPCD to classify mastery patterns according to the distances between the observed response and the ideal response, determined using the Q-matrix of the test. However, NPCD has two limitations. First, more than one ideal response may have the "same and shortest (closest) distance" to the observed response. Therefore, the estimated mastery pattern is randomly selected from the mastery pattern candidates. To avoid this, the NPCD uses a weighted Hamming distance instead of the Hamming distance. Nevertheless, if one considers the weighted Hamming distance, then the variances of items should be estimated by students' observed responses. This is the second problem, and generally, NPCD with the original weighted Hamming distance is not a real parametric-free model.

In this study, the variances of the ideal responses were used instead of the variances of the observed responses of students. We also used the EAP concept to combine the mastery pattern candidates. The normalized inverse distance from the observed response to the ideal responses was calculated, and the probabilities of the attributes of a student were computed by combining the mastery patterns through the corresponding normalized inverse distances. The proposed ideal response-based weighted Hamming distance was also applied to compute the similarities between the observed response and ideal responses. Hence, the proposed methods were called NWCD models. We used two NWCD models, namely NWSD to classify students' mastery skills and NWBD to classify students' existing bugs.

The experimental results on both simulated datasets indicate that NWSD obtains the best classification accuracy compared with DINA, G-DINA, and NPCD. Moreover, NWBD outperforms Bug-DINO for classifying students' existing bugs. In addition, NWCD models are also appropriate for use in cases with a small class. Therefore, the proposed NWCD models can overcome the two drawbacks of NPCD simultaneously. Further, both NWSD and NWBD are suitable for estimating just one student's mastery/bug patterns.

The purpose of this manuscript is for application in classroom teaching and some small units. Hence, in practice, the number of skills is not very large. Suppose the number of skills is too large; for example, applying these methods to an online Adaptive Learning Platform [48] from K1–K12. In that case, the computation time may increase substantially, limiting immediate feedback. Therefore, in the future, we will try to combine the learning space concept and nonparametric CDMs to extend them as cross-grade adaptive learning algorithms and solve huge skill problems.

**Author Contributions:** Conceptualization, C.-H.L.; methodology, C.-H.L. and P.-J.H.; software, C.-H.L. and P.-J.H.; validation, P.-J.H. and Y.-J.J.; data curation, Y.-J.J.; writing—original draft preparation, C.-H.L.; writing—review and editing, C.-H.L., P.-J.H. and Y.-J.J.; project administration, C.-H.L. All authors have read and agreed to the published version of the manuscript.

**Funding:** This work was supported by the Ministry of Science and Technology, Taiwan, under Grant MOST 103-2511-S-142-012-MY3, 109-2511-H-142-004-, and 110-2511-H-142-006-MY2.

**Institutional Review Board Statement:** The study protocol was approved by the Central Reginal Research Ethics Committee (CRREC) of the China Medical University (protocol code N/A/CRREC-109-070, approved on 15 June 2020).

**Informed Consent Statement:** Informed consent was obtained from all subjects involved in the study.

**Data Availability Statement:** Data are not publicly available but may be made available on request from the corresponding author.

**Acknowledgments:** We thank the participants of this study. We are also grateful to the team of Bor-Chen Kuo for providing the Taiwanese CPS website.

**Conflicts of Interest:** The authors declare no conflict of interest.

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
