# Peer review of "A Nonparametric Weighted Cognitive Diagnosis Model and Its Application on Remedial Instruction in a Small-Class Situation"

_sustainability, doi:10.3390/su14105773_

Round 1

Reviewer 1 Report

The author proposes two cognitive diagnostic models NWSD and NWBD, which are used in primary school classroom teaching practice. The experiments are more detailed, but there are still the following problems.

  1. The models used by the author for comparison are too old and lack the comparison of related work models in recent years. Therefore, the model proposed by the author is lacking in persuasiveness.
  2.  Cognitive diagnosis has also published a lot of new work in recent years. The author's citations are too old and need to be supplemented.
  3. However, the NWSD and NWBD supplementary courses mentioned in the article are superior to the traditional group supplementary courses, which can properly summarize the research status of supplementary courses, and focus on describing the advantages of their supplementary courses; the conclusion part also suggests an appropriate outlook for their next work;
  4. In the abstract section, the existing problems in this field are not indicated, why the author chooses two non-parametric weighted cognitive diagnostic models NWSD and NWBD.
  5. The description in 3.3 (A Nonparametric CDM Website) is too little and can be discussed in more detail. Figure 1 is blurry and should increase the clarity of Figure 1.

Author Response

Point 1: The author proposes two cognitive diagnostic models NWSD and NWBD, which are used in primary school classroom teaching practice. The experiments are more detailed, but there are still the following problems.

Response 1: Thank you very much for your kind words about our paper. We have already revised this paper to make it better.

Point 2: The models used by the author for comparison are too old and lack the comparison of related work models in recent years. Therefore, the model proposed by the author is lacking in persuasiveness.

Response 2: Thanks for these suggestions. As the reviewer mentioned, we want to implement the proposed method in primary school classroom teachings. Hence, we choose some robust models such as DINA, G-DINA, and NPCD. For example, G-DINA has recently been applied to other research areas [27, 49]. In the future, we will try to compare the proposed method with other new CDMs. We have added the further reference [49] and briefly described it in the revision on page 2.

27. de la Torre, J., van der Ark, L. A., & Rossi, G. Analysis of clinical data from a cognitive diagnosis modeling framework. Measurement and Evaluation in Counseling and Development, 2018, 51(4), 281-296.

49. Liang, Q., de la Torre, J., & Law, N. Do background characteristics matter in Children's mastery of digital literacy? A cognitive diagnosis model analysis. Computers in Human Behavior2021, 122, 106850.

Point 3: Cognitive diagnosis has also published a lot of new work in recent years. The author's citations are too old and need to be supplemented.

Response 3: Thanks for these suggestions. We have cited some new work in the revision.

Point 4: However, the NWSD and NWBD supplementary courses mentioned in the article are superior to the traditional group supplementary courses, which can properly summarize the research status of supplementary courses, and focus on describing the advantages of their supplementary courses; the conclusion part also suggests an appropriate outlook for their next work; In the abstract section, the existing problems in this field are not indicated, why the author chooses two non-parametric weighted cognitive diagnostic models NWSD and NWBD.

Response 4: Thanks for these suggestions. Thanks for these suggestions. In the abstract, we have added the existing problems and reasons why we proposed and chose two non-parametric weighted cognitive diagnostic models, NWSD and NWBD.

Point 5: The description in 3.3 (A Nonparametric CDM Website) is too little and can be discussed in more detail. Figure 1 is blurry and should increase the clarity of Figure 1.

Response 5: Thanks for these suggestions. More description in 3.3 can help authors to apply the method easily. Hence, we have added the four steps to obtain the students' mastery files on page 5. Moreover, we use a clarity figure instead of the original Figure 1 in the previous manuscript.

Reviewer 2 Report

Firstly, I would like to congratulate the authors for their idea and effort. The article is nice and also provide great views on the topic of cognition in regard to education.

Except few small corrections such as: in formula 1 I guess k needs to take value from 1 to K and not K’ or in formula 8 a 2’ appears, there are not so many things needed to be done before publishing. Also please add some explanations for the arguments on line 102 – they seem similar.

Author Response

Point 1: Firstly, I would like to congratulate the authors for their idea and effort. The article is nice and also provide great views on the topic of cognition in regard to education.

Response 1: Thank you very much for your kind words about our paper. We have already revised this paper to make it better.

Point 2: Except few small corrections such as: in formula 1 I guess k needs to take value from 1 to K and not K’ or in formula 8 a 2’ appears, there are not so many things needed to be done before publishing. Also please add some explanations for the arguments on line 102 – they seem similar.

Response 2: Many thanks for indicating these typo errors. We have rewritten the arguments on line 102 shown in the previous version.

Reviewer 3 Report

See the attachment for more details.

Author Response

Point 1: This paper proposes two nonparametric weighted cognitive diagnosis models, NWSD and NWBD, for classifying mastery skills and lack bugs, respectively. These two methods improve previous models which requires a lot of parameter estimation. To estimate the parameters of the model, it is necessary to collect a large number of examinees'answer information, so it is not suitable for small-class teaching. The NWCD methods propose in this paper is similar to the NPCD method, which uses weighted Hamming distance to compare the ideal answer situations and diagnose the current skill mastery state of students. Using hamming distance avoids a lot of parameter estimation. Diagnosing the students'skill mastery is the same as the students'misconceptions model. In this paper, the effectiveness of the proposed method is verified by using simulated data sets. Then experiments are carried out on real data sets to verify the practicability of small-class teaching. General speaking, this paper has a quality for publication after a small revision.

Response 1: Thank you very much for your kind words about our paper. We have already revised this paper to make it better.

Point 2: In view of this article, I put forward the following questions:

For the introduction, cognitive diagnosis has also been studied in the field of machine learning More reviews would be better, such as "Meta-knowledge dictionary learning on 1-bit response data for student knowledge diagnosis[J]. Knowledge-Based Systems, 2020, 205: 106290." and "Knowledge Tracing: A Survey[J]. arXiv preprint arXiv:2201.06953, 2022."

Response 2: Many thanks for the suggestions. These two references are pretty good, and we have cited and introduced them in the revision on page 1. We agree with the review that the deep-learning-based algorithm can be applied to CDMs. In 2020, we also proposed a deep-learning-based CDM through the autoencoder architecture shown in the following. However, the paper is written in Mandarin, so we didn't cite it. Although the proposed method is not based on network architecture,  the ideal is similar to the weighted k nearest neighbor classifier and weighted mean concept in the machine learning category. It is easier to understand than deep-learning-based CDMs. Moreover, the NWSD and NWBD only used ideal responses and are without parameters. Hence, we expect the diagnosis speed is faster than deep-learning-based CDMs.

           Furthermore, one paper [5] we have cited in the previous manuscript is about computerized dynamic adaptive tests. The idea is based on the knowledge structure of concepts. In our research plan, the next step is to combine the learning space concept and nonparametric CDMs to extend them as cross-grade adaptive learning algorithms.

Li, C.-H., Hsieh, P.-J., & Liu, Z.-Y. A Cognitive Diagnosis Model Based on Partially Connected Neural Network and Its Application on Small-Class Teaching. Psychological Testing, 2020, 67(2), 145-166.

5. Wu, H. M., Kuo, B. C., & Wang, S. C. Computerized dynamic adaptive tests with immediately individualized feedback for primary school mathematics learning. Journal of Educational Technology & Society, 2017, 20(1), 61-72.

Point 3: For models, in Eq.(11), the distance calculation seems to require all the ideal response. So, how about the case where the number of skills is very large? What about the computational efficiency of the algorithm? Giving the algorithm complex analysis will be better.

Response 3: Many thanks for indicating that. The purpose of the manuscript is to apply in classroom teaching and for some small units. Hence, in practice, the number of skills is not very large. However, our next step is to apply the proposed in the online Adaptive Learning Platform [44] from K1-K12. The proposed method will face a problem: the number of skills is enormous, the reviewer mentioned. Recently, we have been trying to combine the learning space concept and nonparametric CDMs to extend them as cross-grade adaptive learning algorithms and solve huge skills problems.

44. Kuo, B. C. (2020). Adaptive Learning. Available online: https://adl.edu.tw/HomePage/home/ (accessed on 18 March 2022).

Point 4: In cognitive diagnosis, do the Q-matrix and m-matrix have to be necessary given by experts? If there is no marker, can we still diagnose students'mastery of skills? The assumption may be required to be definified.

Response 4: Many thanks for the suggestion. We agree with the reviewer. Many ways can be applied to diagnose students' mastery of skills, such as item response theory, knowledge structure-based algorithm, and deep-learning-based tracing algorithms. However, in most CDM models, the Q-matrix is commonly used and can be thought of as experts' opinions.

We have revised the sentence to decrease the misunderstanding of readers. CDMs can be used to provide diagnostic conclusions about examinees’ mastery skills. Some of them are according to a given Q-matrix of a test and their responses [28, 31-32, 34].

Point 5: For the experiments, the datasets used for evaluation are too small? How to guarantee the statistical significance? Does the public datasets could be used, e.g. the Assistment dataset? But, basically, the test looks well, especially for the experimental group and control group

Response 5: Many thanks for the suggestion. Due to the revision deadline, we will try the public data, the Assistant dataset, in the future.

Point 6: There are two Section 6 might cased by typos. The authors could disscuss more on the limition on the number of skills and the applications. Besides, the related studies on education outcomes could also be discussed, e.g., "Predicting and Understanding Student Learning Performance Using Multi-source Sparse Attention Convolutional Neural Networks[J], IEEE Transactions on Big Data, 2021."

Response 6: Many thanks for indicating the typos. In addition, we have added the following limitation at the end of the conclusion. The purpose of the manuscript is to apply in classroom teaching and for some small units. Hence, in practice, the number of skills is not very large. Suppose the number of skills is too large; for example, applying these methods in the online Adaptive Learning Platform [44] from K1-K12. In that case, the computation time may increase a lot, and cannot immediately provide the feedback. Therefore, in the future, we will try to combine the learning space concept and nonparametric CDMs to extend them as cross-grade adaptive learning algorithms and solve huge skills problems.
